# Configuring Agentic AI Coding Tools: An Exploratory Study

## Abstract

Agentic AI coding tools with autonomous capabilities beyond conversational content generation increasingly automate repetitive and time-consuming software development tasks. Developers can configure these tools through versioned repository-level artifacts such as Markdown and JSON files. In this paper, we present a systematic analysis of configuration mechanisms for agentic AI coding tools, covering Claude Code, GitHub Copilot, Cursor, Gemini, and Codex. We identify eight configuration mechanisms and, in an empirical study of 2,926 GitHub repositories, examine whether and how they are adopted. We then explore CONTEXT FILES, SKILLS, and SUBAGENTS, that is, three mechanisms available across tools, in more detail. Our findings reveal three trends. First, CONTEXT FILES dominate the configuration landscape and are often the sole mechanism in a repository, with AGENTS.md emerging as an interoperable standard across tools. Second, advanced mechanisms such as SKILLS and SUBAGENTS are only shallowly adopted: most repositories define only one or two artifacts, and SKILLS predominantly rely on static instructions rather than executable workflows. Third, distinct configuration cultures are forming around different tools, with Claude Code users employing the broadest range of mechanisms. These findings establish an empirical baseline for longitudinal and experimental research on how configuration strategies evolve and affect agent performance as agentic AI coding tools mature.

## CCS Concepts

• **Software and its engineering**;

## Keywords

Software Engineering, Generative AI, AI Agents, Configuration

**ACM Reference Format:**
Anonymous Author(s). 2026. Configuring Agentic AI Coding Tools: An Exploratory Study. In *3rd ACM International Conference on AI-powered Software (AIware 2026), July 6–7, 2026, Montreal, Canada.* ACM, New York, NY, USA, 9 pages. https://doi.org/10.1145/XXXXXXX.XXXXXXX

## 1 Introduction

Agentic AI coding tools based on large language models (LLMs) [24, 26], automate time-consuming and repetitive tasks, such as generating and editing code, tests, and documentation. The agentic capabilities of these tools go beyond those of purely reactive conversational assistants. Instead of just following user commands, these tools take initiative and act proactively to accomplish defined objectives [2]. They can autonomously interact with development environments and use project artifacts, external data, and command-line tools to perform actions with reduced human interaction [28]. Certain functionality can be delegated to "tools" and "(sub)agents." Tools are capabilities with a specific bounded function (e.g., `run_python(code)` or `search_web(query)`).[1] They are deterministic and invoked by the underlying model or agents. (Sub)agents are goal-directed action-taking components. They interpret a user goal, break it down into substeps, choose appropriate tools, execute actions, observe results, and potentially adjust their plan for the next iteration of this "agent loop." Initially, agentic AI coding tools implemented one central agent loop steered by a foundational model such as Anthropic's Opus or OpenAI' GPT. Example tools include Claude Code and OpenAI Codex. Soon, conversational tools such as GitHub Copilot and Cursor offered similar capabilities in an "agent mode." Over the months preceding the writing of this paper, tool vendors implemented extension and configuration mechanisms that allow developers, for example, to write their own (sub)agents that operate in parallel to the central agent loop.

We define *configuration mechanisms* as means by which developers can tailor tool and agent behavior to a project or workflow (e.g., by providing contextual information in a file or by specifying dedicated subagents). A *configuration artifact* is the tangible specification of that mechanism (e.g., the actual Markdown file with the contextual information). For example, files like `AGENTS.md` or `CLAUDE.md` act as "READMEs for agents" with context-specific information about build commands, coding conventions, and rules for CI/CD pipelines [16, 18]. Configuration artifacts are version-controlled, inspectable, and collaboratively maintained. With the increased availability and diversity of agentic tools, the number of configuration mechanisms and related artifacts has also increased.

To understand the current landscape of agentic tools' configuration mechanisms and their use in open-source software (OSS), we address the following **research questions:**

**RQ1** *What configuration mechanisms do agentic coding tools offer?*
**RQ2** *Which mechanisms are adopted in OSS repositories?*
**RQ3** *How are configuration mechanisms adopted?*

We consider configuration mechanisms that are captured in configuration artifacts to be consumed by agentic tools, are repository-versioned for collaborative maintenance, and include instructions intended to customize the behavior of AI coding tools.

With this paper, we provide the following **contributions:** (1) We systematically documented eight configuration mechanisms: CONTEXT FILES, SKILLS, SUBAGENTS, COMMANDS, RULES, SETTINGS, HOOKS, and MCP servers. We identified these from the documentation of Claude Code, GitHub Copilot, Cursor CLI, Gemini CLI and Codex CLI. Some mechanisms are available in all tools; others are specific to particular tools. (2) We analyzed the adoption of these configuration mechanisms in 2,926 GitHub repositories. CONTEXT FILES (Markdown files that provide contextual project information) dominated and are often the sole configuration mechanism. Claude Code users apply the broadest range of configuration mechanisms.

---

[1]We distinguish such "tools" from full-fledged "agentic tools" (e.g., Claude Code).

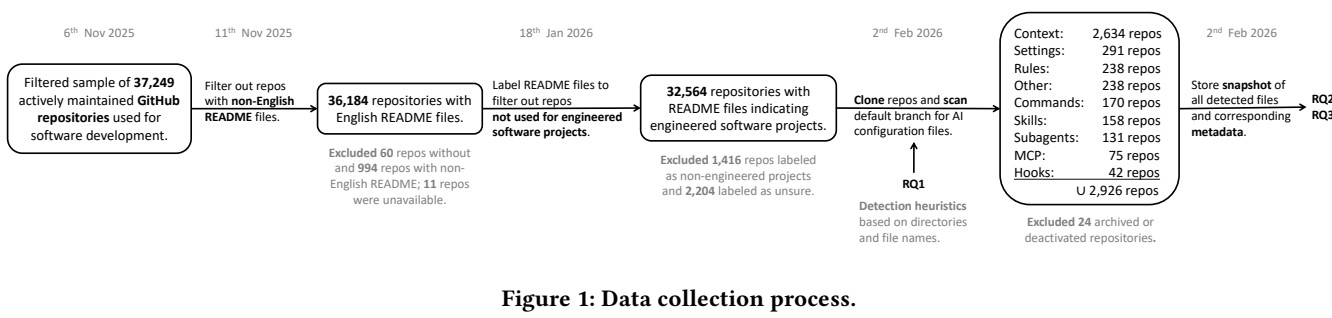

**Figure 1: Data collection process.**

(3) We analyzed the adoption of CONTEXT FILES, SKILLS (bundles of information and tools), and SUBAGENTS (specialized agents with their own model context), in more detail. Three CONTEXT FILE formats showed the most dynamic development, with AGENTS.md emerging as a standard. Although SKILLS and SUBAGENTS can be defined for a wide range of purposes, most repositories that adopt them define only one or two artifacts. Moreover, SKILLS primarily rely on static resources rather than executable scripts to extend tool and agent behaviour.

## 2 Related Work

Related work has examined how repository-level context files, prompts, and structured context mechanisms shape agent behavior, efficiency, and integration into development workflows. Although several studies have investigated individual configuration artifacts, most notably repository-level context files such as AGENTS.md, prior work has typically focused on single artifact types in isolation. There has been no systematic examination of the broader configuration mechanisms to customize agentic behavior. Independent of tool architecture, context engineering allows developers to design, structure, and supply task- and project-specific information to guide the behavior of tools [9, 11, 15, 20, 22]. Empirical work has mainly focused on repository-maintained context files, such as AGENTS.md. Mohsenimofidi et al. [16] analyzed the emergence of context engineering practices in open-source projects and showed that repository-level instruction files serve as persistent configuration mechanisms that encode architectural constraints, build commands, and workflow conventions. Complementing this work, Chatlatanagulchai et al. [6] found that context files are actively maintained, structurally consistent, and focus predominantly on functional development instructions such as build and testing procedures. Santos et al. [21] analyzed Claude Code projects and how configuration artifacts structure agent interactions in repositories.

Beyond structural characterization, recent work has begun to evaluate the impact of such artifacts. Lulla et al. [13] conducted a controlled study comparing agent executions with and without an AGENTS.md file and reported lower runtime and token consumption while maintaining comparable task completion behavior.

In other work, Villamizar et al. [27] argued that prompts should be treated as software engineering artifacts and outlined a research agenda to understand their evolution, reuse, and governance. Beyond static prompts, Zhang et al. [30] introduced a framework that treats contexts as evolving instructions and addresses issues such as brevity bias and context collapse through incremental updates.

Recent work further generalized this perspective by formalizing context as a programmable abstraction. Zhang and Wang [31] introduced Monadic Context Engineering, modeling context construction and transformation through monadic composition. McMillan [14] evaluated structured file-native context schemas and showed how format and multi-file organization influence retrieval accuracy and downstream agent performance. Ye et al. [29] proposed Meta Context Engineering, a framework in which agentic skills and context artifacts co-evolve via evolutionary search.

However, a cross-tool and cross-agent perspective mapping configuration mechanisms and studying their adoption patterns across repositories is missing. Our work addresses this gap.

## 3 Data Collection and Analysis

We collected OSS projects hosted on GitHub. Since GitHub hosts a variety of content, we selected repositories belonging to "engineered" software projects [17]. We adopted and updated a selection approach used in prior work, starting with the SEART GitHub search tool [8, 16, 23]. We selected non-fork repositories with at least two contributors and a license that were created before 1 January 2024 with commits since 1 June 2025. We then excluded archived, disabled, or locked repositories We applied a licensing filter and a language filter based on popularity, selecting Python, TypeScript, JavaScript, Go, Java, C++, Rust, PHP, C#, and C, and applied filters based on commit and watcher count based on prior research [16]. This resulted in a final sample of 37,249 repositories.

Figure 1 outlines our data collection pipeline for these repositories. We cloned them and searched their default branch for a README file. We excluded 11 repositories that became unavailable in the meantime, 60 repositories without a README file, and then used the lingua-language-detector Python library to detect the language of files we found. This resulted in 994 repositories with non-English README files being excluded.

For the remaining 36,184 repositories, we developed a classification pipeline that uses the GPT-5.2 model to determine, based on the content of a README file, whether the software project fulfills our definition of an "engineered" project. We based our definition on Munaiah et al.'s paper [17]. An engineered project is a project that shows clear evidence of software engineering practices (e.g., it includes a clearly stated purpose of a problem that the software solves; it provides installation and deployment instructions, developer and user documentation, testing, CI/CD, or quality-assurance-related information). There should also be evidence for maintenance. This definition was part of our prompt, together with the README of each repository. In our supplementary material, we share the final

**Table 1: Overview of repository-level configuration mechanisms across agentic AI coding tools. Each cell lists the repository file(s) or directory implementing that mechanism; "–" indicates that the mechanism is not available.**

| Mechanism | Description | Claude | Copilot | Codex | Cursor | Gemini |
|---|---|---|---|---|---|---|
| CONTEXT FILES | Markdown file providing persistent context loaded every session. | `CLAUDE.md` | `.github/ { copilot-instructions.md | instructions/*.md }`[a] | `AGENTS.md`, `AGENTS.override.md` | `AGENTS.md`, `.cursorrules`[c] | `GEMINI.md` |
| SETTINGS | JSON/TOML config for project-level tool behavior. | `.claude/ settings (local)?.json` | _[b] | `.codex/ config.toml` | `.cursor/ cli.json` | `.gemini/{ settings.json | config.yaml }` |
| SKILLS | Reusable knowledge and invocable workflows. | `.claude/skills/` | `.github/skills/` | `.codex/skills/` | `.cursor/skills/` | `.gemini/skills/` |
| SUBAGENTS | Specialized agents running in an isolated context. | `.claude/agents/` | `.github/agents/` | – | `.cursor/agents/` | – |
| COMMANDS | User-triggered shortcuts for predefined prompts. | `.claude/ commands/` | – | – | `.cursor/ commands/` | `.gemini/ commands/` |
| HOOKS | Scripts executed at specific agent lifecycle points. | `.claude/ settings.json` | `.github/hooks/*.json` | – | `.cursor/ hooks.json` | `.gemini/ settings.json` |
| RULES | System-level instructions to control agent behavior. | – | – | `.codex/rules/` | `.cursor/rules/` | – |
| MCP | External tool or data connections via the Model Context Protocol. | `.mcp.json` | _[b] | `.codex/ config.toml` | `.cursor/ mcp.json` | `.gemini/ settings.json` |

[a] Copilot also supports `CLAUDE.md`, `AGENTS.md`, and `GEMINI.md`; [b] Configured via the GitHub web interface, not via files in the project repository; [c] Cursor deprecated `.cursorrules` and now suggests using `AGENTS.md` instead.

**Table 2: Release dates of agentic AI coding tools and repositories ($n = 2,926$; one repository can use multiple tools).**

| Agentic Tool | Release (Month/Year) | #Repositories |
|---|---|---|
| Claude Code | 02/2025 (CLI & agents since release) | 1,310 |
| GitHub Copilot | 10/2021 (Release) 02/2025 (Copilot Agent Mode) 09/2025 (Copilot CLI) | 958 |
| Codex CLI | 04/2025 (CLI & agents since release) | 558 |
| Cursor CLI | 03/2023 (Release) 06/2025 (Cursor Agents) 08/2025 (Cursor CLI) | 355 |
| Gemini CLI | 02/2024 (Release) 05/2025 (Gemini Agent Mode) 06/2025 (Gemini CLI) | 186 |

**Table 3: Repository metadata comparison by agentic tool. Cells show median, IQR, and Cliff's $\delta$ for tool-adopting repositories. Significant differences (BH-adjusted) are in bold.**

| Agentic tool | Age (years) | Contrib. | Commits | Size (KB) |
|---|---|---|---|---|
| All ($n$=2,926) | 6.7 (4.3–9.4) | 42 (19–105) | 2,106 (965–5,064) | 40k (9,498–132k) |
| Claude ($n$=1,310) | **6.2***** (4.1–9.2) $\delta$=-0.08 | 40 (19–111) $\delta$=0.00 | 2,234 (995–5,268) $\delta$=0.03 | 42k (10k–151k) $\delta$=0.04 |
| Gemini ($n$=186) | 6.8 (4.7–9.5) $\delta$=0.03 | **66***** (32–144) $\delta$=0.21 | **3,322***** (1,290–9,776) $\delta$=0.21 | **58k**** (15k–193k) $\delta$=0.12 |
| Codex ($n$=568) | 6.9 (4.2–9.3) $\delta$=0.01 | 43 (19–89) $\delta$=-0.02 | **1,809**** (877–4,058) $\delta$=-0.10 | **34k**** (8,696–94k) $\delta$=-0.08 |
| Copilot ($n$=958) | **7.1***** (5.0–9.7) $\delta$=0.11 | 42 (19–113) $\delta$=0.02 | 2,200 (1,000–5,591) $\delta$=0.04 | 44k (9,858–143k) $\delta$=0.03 |
| Cursor ($n$=355) | **5.5***** (3.4–7.9) $\delta$=-0.19 | **54**** (23–120) $\delta$=0.10 | **2,780***** (1,264–5,968) $\delta$=0.14 | **75k***** (21k–201k) $\delta$=0.22 |

*Note:* Mann–Whitney U test (two-sided) with Benjamini–Hochberg FDR correction (20 comparisons). Effect sizes: Cliff's delta. $^*$ $p < .05$; $^{**}$ $p < .01$; $^{***}$ $p < .001$ (adjusted).

prompt and previous iterations of the prompt, which we tested with subsets of the data, as well as the exact configuration for labeling repositories as "engineered" or not (we kept OpenAI's default values for all parameters) [5]. The classification pipeline labeled 32,564 repositories as engineered software projects, was 'unsure' about 2,204 repositories, and excluded 1,416 repositories. For each label category, we spot-checked randomly selected samples, both during the iterative development of the prompts and for the final sample.

We then cloned the remaining 32,564 repositories and applied heuristics based on file names and file paths to detect the usage of AI coding tools and configuration mechanisms (see Table 1). These heuristics are based on the answers to **RQ1** and are briefly discussed in Section 4. This resulted in 2,926 repositories that use one or more AI coding tools. Our data collection and analysis scripts and the analyzed data are available online [5].

## 4 Configuration Mechanisms (RQ1)

To determine the set of configuration mechanisms for agentic AI coding tools, we followed two steps. We first selected five agentic AI coding tools based on the *2025 Stack Overflow Developer Survey* [25]. The tools represent the four most popular AI tools among the surveyed developers. We added Codex instead of ChatGPT as Codex is an agentic tool by the same vendor using the same models. We added Cursor because it has recently been studied in software

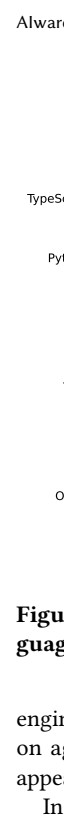

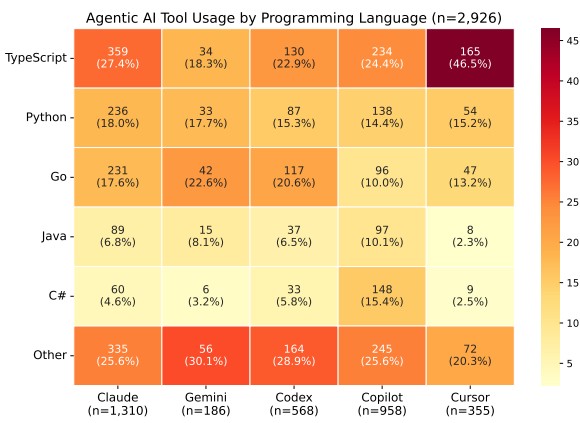

**Figure 2: Adoption of agentic tools per programming language; % relative to #repositories using a certain tool.**

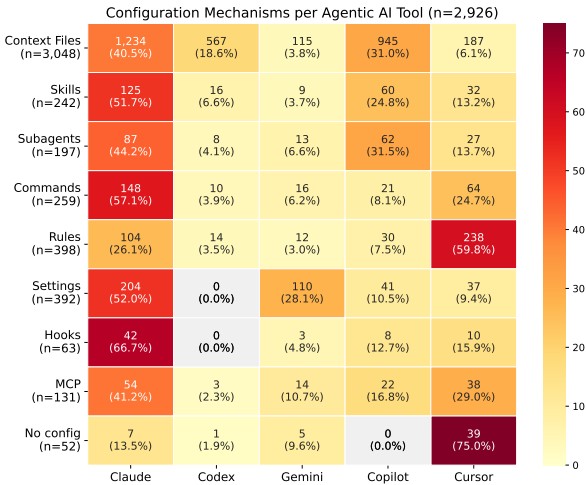

**Figure 3: Usage of configuration mechanisms across agentic tools. % relative to #repositories using a certain mechanism.**

engineering [10, 12]. Table 2 lists the five tools. Our study focuses on agentic tools with a command-line interface (CLI) that first appeared in 2025. For some tools, prior non-agentic versions exist.

In the second step, one author systematically reviewed the online documentation of each tool, documenting configuration mechanisms together with repository-level files and directories that might indicate their usage. This was then cross-checked by two other authors and discussed among all.

As part of **RQ1** and to "operationalize" configuration mechanisms and their occurrence in repositories, we then used that information to develop heuristics to identify whether tools are used in a repository (see Table 1). Note that for GitHub Copilot, Cursor, and Gemini, files and directories indicating their use apply both to their conversational and agentic interfaces. For example, a `.claude` folder or a `CLAUDE.md` file indicate usage of Claude Code. In addition, we developed strategies to determine whether configuration mechanisms were used. For example, if a directory `.claude/agents/` with Markdown files is present, this indicates usage of Subagents. All documents and the Python scripts that implement our matching strategy are part of the supplementary material [5].

> **Key insights for RQ1:**
> - Qualitative analysis of the documentation of agentic tools revealed eight configuration mechanisms.
> - There is convergence for some mechanisms across tools (e.g., all five support Context Files and Skills), yet no single tool implements all eight mechanisms, suggesting the ecosystem is still maturing, standards are developing.

## 5 Adoption of Configuration Mechanisms (RQ2)

To understand how developers adopt configuration mechanisms in agentic AI coding tools, we analyze the presence and co-occurrence of configuration artifacts across repositories.

### 5.1 Characterization of Repositories in Dataset

Before analyzing adoption, we characterize the 2,926 repositories in our dataset to contextualize the configuration mechanisms of agentic AI coding tools.

Most repositories adopted a single tool. Among repositories with multiple tools, Claude appeared most frequently with others. The most common combination was Claude and Copilot ($n = 128$). Overlaps of three or more tools were rare. In our initial sample, the top five main programming languages per repository ($n = 36, 194$) were Python (8,133; 22.5%); TypeScript (4,999; 13.8%); Java (3,813; 10.5%); Go (3,786; 10.5%); and JavaScript (3,709; 10.3%). Interestingly, this order is slightly different for repositories that use agentic coding tools ($n = 2,926$, see Figure 2) where TypeScript and Go were more prominent and C# replaces JavaScript: TypeScript (922, 31.5%); Python (548, 18.7%); Go (533, 18.2%); C# (256, 8.8%); Java (246, 8.4%). TypeScript is the most common primary language for repositories using Claude, Codex, Copilot, and Cursor. Usage in Java and C# repositories is lower for all tools but Copilot. As shown in Table 3, repositories using Cursor are younger compared to those using other tools. Repositories associated with Gemini exhibit larger contributor counts and commit volumes, compared to those associated with other tools. Cursor repositories are the largest in terms of kilobytes of source code.

### 5.2 Distribution of Configuration Mechanisms

Figure 3 shows the distribution of configuration mechanisms. Across repositories, Context Files (with artifacts such as `CLAUDE.md` or `AGENTS.md` files) were the most frequently adopted configuration mechanism. In contrast, other configuration mechanisms, including Rules, Settings, Commands, Skills, and Subagents, were substantially less prevalent. Each of these mechanisms appeared in fewer than 20% of the repositories. Rules are primarily concentrated in repositories using Cursor, since Cursor was one of the first to introduce Rules [12].

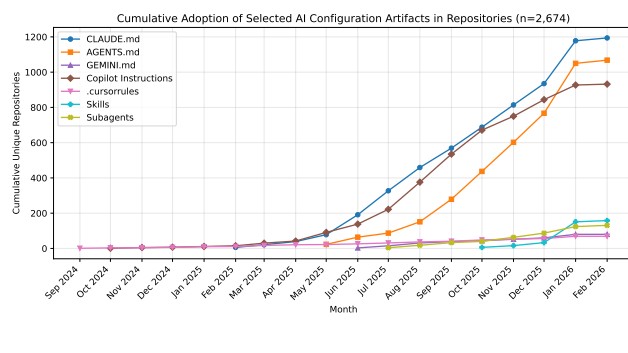

**Figure 4: Cumulative adoption of selected configuration artifacts for agentic tools. Copilot Instructions include the two different artifact types for Context Files, see Table 1.**

Most repositories adopt a limited subset of available configuration mechanisms. In many cases, repositories include only a single Context File artifact. The adoption of multiple non-context file configuration mechanisms within the same repository remains comparatively rare. These results suggest that configuration usage is currently characterized by one dominant baseline configuration (i.e., Context Files).

*5.2.1 Co-occurrence of Configuration Mechanisms.* Several configuration mechanisms are frequently adopted together. Some of these correlations can be explained by shared configuration artifacts. For example, Settings and Hooks have a strong positive correlation ($\rho = 0.52$), because hooks are usually defined in Settings files. However, Settings and Hooks also both co-occur with Skills ($\rho = 0.23$ and $\rho = 0.24$, respectively), which are defined and configured differently. Subagents also show positive correlations with multiple mechanisms, including Settings, Skills, and MCP. Other configuration mechanisms appear less frequently in combination. Context Files are negatively correlated with Rules ($\rho = -0.36$) and Commands ($\rho = -0.14$), which can again attributed to how they are defined and which tools first introduced the mechanisms. Overall, configuration mechanism adoption across repositories is characterized by recurring combinations of mechanisms rather than uniform usage. Patters of co-occurrence might be impacted by the tools and configuration mechanisms they support (e.g., Claude, the dominating tool, does not support Rules).

*5.2.2 Adoption of Configuration Mechanisms over Time.* In Figure 4, we show the adoption of configuration mechanisms over time.

The figure shows individual artifacts that together form the mechanism Context Files, plus Skills and Subagents which have a uniform format.

Context Files (CLAUDE.md, AGENTS.md, copilot-instructions.md) clearly dominate and increase continuously, while Skills and Subagents experienced a comparatively slow growth. As the figure shows, .cursorrules and copilot-instructions.md started being introduced in 2024. Although Cursor and also Copilot were originally released in 2023 and 2021, respectively, agentic capabilities were only introduced in 2025 (see Table 2). The adoption of copilot-insturctions.md has increased since then.

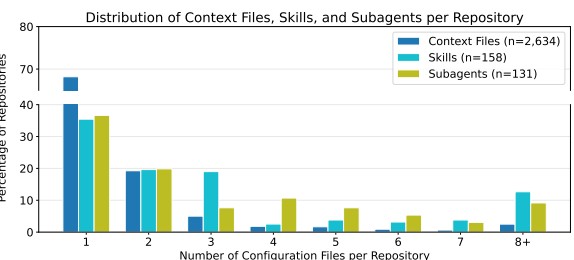

**Figure 5: Configuration mechanism count per repository.**

> **Key insights for RQ2:**
> - Context Files are the dominant and often sole configuration mechanism. Advanced configuration mechanisms (e.g., Hooks, MCP) are adopted less frequently.
> - Adoption varies: Claude has a broad configuration footprint, while Cursor emphasizes rule-based mechanisms.
> - Although certain correlations between configuration mechanisms exist, the landscape still appears in flux.

## 6 Details of Configuration Mechanisms (RQ3)

In this section, we analyze the use of three configuration mechanisms in more detail. We selected Context Files and Skills because they are supported by all tools, as discussed in Section 5, and Subagents because they follow a similar format as Skills.

### 6.1 Configuration Mechanism: Context Files

Context Files are Markdown files that provide a central machine-readable source of contextual information about the repository in which they are used. AGENTS.md was introduced by OpenAI and now serves as an open tool-agnostic convention that is increasingly supported by various tools [16]. We analyzed the presence and co-occurrence of Context Files across repositories. We identified 4,860 Context Files across 2,634 of the 2,926 repositories in our sample. The histogram in Figure 5 shows the distribution of Context Files. Most repositories that do use Context Files include one or two of such files.

CLAUDE.md emerges as the dominant file type with 1,661 (34.2%), followed closely by AGENTS.md and copilot-instructions.md with 1,572 (32.3%) and 1,344 (27.7%) files, respectively. GEMINI.md (159 files, 3.3%) and .cursorrules (73 files, 1.5%) are rare. Note that .cursorrules are now deprecated; Cursor suggests using AGENTS.md instead. Cursor rules are different and are stored in .cursor/rules/. We also investigated the repository-level adoption of the Context Files, where CLAUDE.md demonstrates the highest adoption rate with 45.40% of repositories (1,195 repos), AGENTS.md follows closely with 40.6% adoption (1,069 repos), and copilot-instructions.md appears in 35.1% of repositories (925 repos). The predominance of CLAUDE.md, AGENTS.md

and copilot-instructions.md, both in terms of their frequency and the number of repositories adopting them, indicates a clear concentration of projects around these files.

We then investigated programming language-specific patterns in Context Files usage across the repositories. We found Context Files in 651 repositories with TypeScript as the main language, 435 with Python, 414 with Go, 210 with Java, 210 with C#, 201 with JavaScript, 134 with Ruby, 181 with Rust, 156 with C++, 123 with PHP, and 53 with C. CLAUDE.md is the most common Context File across languages, except for Java, C#, and C++, where copilot-instructions.md emerges as the dominant type.

Figure 6 shows the creation order of the different Context Files. Most repositories only used one type of Context Files. These repositories are not captured in this figure. AGENTS.md and CLAUDE.md are typically created first, and if CLAUDE.md existed first, then AGENTS.md was commonly created afterward. A reason for this could be that Claude Code is the most popular agentic AI coding tool, but does not yet support the emerging standard AGENTS.md [4]. Interestingly, repositories that started with copilot-instructions.md then often created CLAUDE.md or AGENTS.md files, even though GitHub Copilot supports all major Context Files types. This, in addition to the overall dominance of CLAUDE.md and AGENTS.md, illustrates that these two files currently serve as the de facto standard.

Finally, we observed that Context Files commonly refer to each other. This is visualized in Figure 7. "Referring to" means that one file, the source, contains a reference to another file, the target, rather than providing their own content. We found three types of references between Context Files:

(1) Direct pointer: Context Files include one line that references files either via their filename (e.g., "AGENTS.md", "@./AGENTS.md"), imperative statements (e.g., "Read @AGENTS.md"), or a Markdown link (e.g., "[anchor text](AGENTS.md)").
(2) Short reference with context: Context Files provide 2–5 lines of what to do with the referenced file, e.g., "*Use instructions from* AGENTS.md *to guide your work.*", "*Always read* AGENTS.md *before answering*", or a Markdown header followed by one of the above.
(3) Brief summary plus reference: Context Files include a header, 1–2 sentences of context, and a reference to another file, e.g., "*# AI Guidelines [...] **Read* @AGENTS.md *for comprehensive guidelines.***"

We found 518 file pairs that reference each other. The file type with most references was CLAUDE.md, with 357 "outgoing" references, followed by AGENTS.md and GEMINI.md. In contrast, the most frequently referenced file was AGENTS.md with 368 "incoming" references, substantially more than CLAUDE.md and copilot-instructions.md. Furthermore, the strongest pattern was the CLAUDE.md to AGENTS.md pair that occurred 311 times. In contrast, when AGENTS.md files acted as a pointer, they most frequently referenced CLAUDE.md files. This again reflects the symbiotic relationship between these two types of Context Files that we also found in the creation order discussed earlier.

## 6.2 Configuration Mechanism: Skills

Skills were introduced by Anthropic and now serve as an open standard for extending the capabilities of agentic AI tools with specialized contextual knowledge and workflows. Skills bundle prompts, tools, and documentation that an agent can invoke on demand. From an implementation perspective, Skills are a directory that contains a SKILL.md file. This file must include a YAML frontmatter with at least a name and description, followed by instructions in the body for how a task should be performed.

We found 601 Skills in 158 repositories. On average, these repositories use 3.8 Skills ($min = 1.0$; $max = 28.0$; $median = 2$). However, as Figure 5 shows, repositories that use Skills typically use fewer than three Skills.

As with Context Files, we also explored whether there are programming language-specific patterns. Figure 2 shows that TypeScript and Python are the dominating programming languages in all repositories that use agentic tools. This distribution is also reflected across projects that use Skills.

The specification of Skills recommends that SKILL.md files should contain fewer than 500 lines. It further recommends moving any detailed material to separate files [1]. We found that of the 601 Skills we identified, only 25 (4%) were longer than 500 lines. That is, most developers follow the recommendation. To provide detailed information and instructions, Skills can utilize different resources, i.e., additional optional files or directories that are loaded by the agents of a tool when required. There are three types of resource:

(1) scripts/ contains executable code that agents can run. Language options include Python, Bash, and JavaScript.
(2) references/ contains documentation that an agent can read when needed. This can include technical references, templates or structured data and other domain-specific files.
(3) assets/ contains static resources, such as templates (document templates, configuration templates), images (diagrams, examples) and data files (lookup tables, schemas).

To obtain the resources used by the different Skills, we scanned all Skills directories in the studied repositories for the occurrence of the relevant resource folders. Of the 601 skills analyzed, the vast majority (501, 83.3%) included no additional resources. Among those that did, the most common pattern was a references/ directory (48 skills, 8.0%), followed by a scripts/ directory (34, 5.7%). Twelve Skills (2.0%) combined both scripts/ and references/, while an assets/ directory appeared in only four Skills (0.7%). The remaining combinations, i.e., references/ with assets/, and all three directories together, each occurred just once (0.2%). In summary, Skills mostly use "static" resources (documentation that the agent reads when needed) rather than "dynamic" resources, such as executable scripts that extend agent behavior.

## 6.3 Configuration Mechanism: Subagents

Subagents are specialized AI agents that agentic AI coding tools or other agents can delegate tasks to. Technically, Subagents are a Markdown file with the same YAML frontmatter structure as Skills. However, Subagents differ in that they operate within their own context window and return the results to the parent agent whereas Skills execute instructions within the calling agent's context [3, 7].

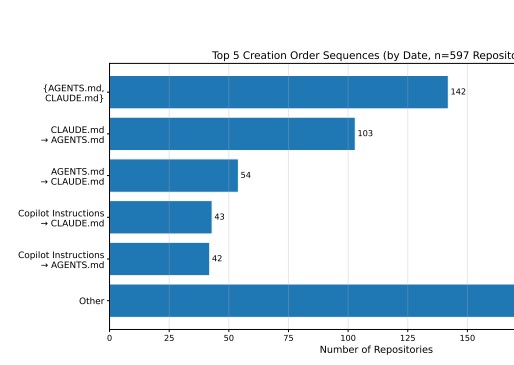

**Figure 6: Creation order of Context Files per repository. Curly braces indicate that files were added on the same day.**

We found 452 Subagents across 131 repositories. On average, these repositories use 3.45 Subagents (*min* = 1.0; *max* = 18.0; *median* = 2). However, Figure 5 indicates that these repositories commonly use fewer than three Subagents. One feature of Claude Code's Subagents is that they can have their own "memory", i.e., a persistent directory that survives across interactions with the agent [3]. This directory can be used to build up knowledge over time, e.g., to store debugging insights. However, we could not find any repositories storing such memory files.

> **Key insights for RQ3:**
> - Context Files: `CLAUDE.md`, `AGENTS.md` and `copilot-instructions.md` are the most common artifacts. Creation and reference patterns revealed that `CLAUDE.md` and `AGENTS.md` serve as de facto standards while `AGENTS.md` emerges as a unifying standard.
> - Skills: Although Skills can be fine-grained and offer various customization features, most repositories that adopt them only define one or two and predominantly utilize static documentation rather than dynamic scripts.
> - Subagents: The usage patterns for Subagents are similar to those of Skills. Future work should study differences in content and use cases in more detail.

## 6.4 Threats to Validity

We organize this section following established guidelines [19].

*Construct validity:* Our heuristics (Table 1) detect the *presence* of configuration artifacts but not whether the corresponding tool is actively used. We partially mitigate this by restricting our sample to actively maintained repositories, but artifact presence remains a proxy for adoption. Additionally, for GitHub Copilot, Cursor, and Gemini, detected files apply to both conversational and agentic modes, so we cannot isolate agentic usage. This limitation does not affect agentic-only mechanisms such as Skills and Subagents, nor the broader `AGENTS.md` trend.

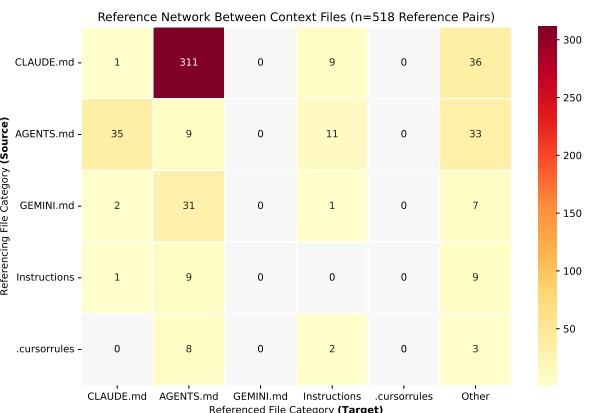

**Figure 7: Reference-only Context Files. Examples of other files include `README.md` and `CONTRIBUTING.md`.**

*Internal validity:* We classified repositories as "engineered" software projects using `GPT-5.2` on their README content, with iterative prompt development and spot-checks to mitigate errors. We used a single labeling run without considering inter-model agreement; 2,204 "unsure" cases (often due to inaccessible linked resources) were excluded. Future work should assess the reliability of this approach by adding alternative models and configurations. Detection heuristics were designed by one author and cross-checked by two others, but rapidly evolving tool ecosystems may introduce conventions not yet captured.

*External validity:* Our study covers only open-source repositories on GitHub; practices in proprietary or enterprise settings may differ. We selected repositories showing established engineering practices, but cannot claim representativeness for closed-source development, nor did we examine variation across application domains. Finally, our findings are a point-in-time snapshot (February 2026) of a rapidly evolving landscape.

## 7 Discussion

Our study offers a first cross-tool snapshot of configuration mechanisms and their artifacts across five agentic AI coding tools and 2,926 repositories. Below, we discuss the implications of our study.

*Standardization around `AGENTS.md`:* Our results reveal a clear trajectory toward `AGENTS.md` as a unifying, tool-agnostic configuration artifact. The creation order and reference patterns point in the same direction: For example, when both files are present, `CLAUDE.md` typically appears first and `AGENTS.md` is added later (Figure 6). `CLAUDE.md` most frequently points to `AGENTS.md` (311 cases), and `AGENTS.md` receives 368 incoming references overall, far more than any other file type. Repositories that began with `copilot-instructions.md` often later introduced `CLAUDE.md` or `AGENTS.md`, despite GitHub Copilot already supporting all major Context File types. Together, these patterns suggest bottom-up convergence around `AGENTS.md` as a cross-tool baseline, driven by developer practice and compatibility needs rather than vendor mandate. At the same time, the layering of multiple context files

within a single repository raises the risk of redundant or conflicting instructions across artifacts. For tool vendors, native support for `AGENTS.md`—as already provided by Cursor and Codex—may become a baseline expectation.

*Shallow adoption despite available depth:* Across all three configuration mechanisms we examined in detail, adoption remains limited. Most repositories include only one or two Context Files (Figure 5), and the median number of Skills and Subagents per repository is two in both cases. Moreover, 83.3% of Skills do not include additional resources, and when resources are present, static documentation (`references/`) is more common than executable scripts (`scripts/`). In practice, Skills, therefore, function primarily as structured text rather than executable workflow bundles. This suggests that configuration is currently used more as documentation than as automation. We also found no evidence of repositories using the persistent memory feature of Claude Code's Subagents.

This gap between available depth and actual usage likely reflects both the novelty of these mechanisms and the effort required to configure them. Claude Code was released only in February 2025, with Skills and Subagents introduced even more recently. Beyond recency, developers may gravitate toward the lowest-friction mechanism (i.e., Context Files) without exploring more advanced options. Defining executable Skills with scripts and structured resources requires additional design and maintenance effort compared to authoring Markdown instructions, which may deter adoption in the absence of clear guidance on best practices. At present, there is little empirical evidence on which configuration strategies are most effective or under which conditions they yield measurable improvements. Future work should therefore assess whether deeper configuration leads to performance gains, extending early evidence on the impact of Context Files [13].

*Distinct tool ecosystems:* Configuration practices vary systematically across tools. Claude Code repositories exhibit the broadest configuration footprint, whereas Cursor projects emphasize Rules and Commands, and Copilot and Codex repositories rarely extend beyond Context Files. These differences likely reflect both the configuration options each tool exposes (Table 1) and the norms emerging within their respective user communities. Tool ecosystems also differ in repository characteristics: Cursor repositories tend to be younger and larger, while Gemini repositories show comparatively higher activity levels (Table 3). An open question is whether these tool-specific configuration cultures will converge as feature sets overlap, or whether distinct patterns of use will persist.

*Practical implications and future directions:* Our systematically compiled overview of eight configuration mechanisms (Table 1) provides a reference for developers seeking to understand the options available to them. Based on our findings, we offer the following concrete observations for practitioners. First, Context Files—particularly `AGENTS.md`—represent the lowest-friction entry point for configuring agentic tools and are already widely adopted. Second, developers who rely on multiple tools should consider maintaining an `AGENTS.md` file as a shared configuration baseline, given its cross-tool support and the reference patterns we observed. When multiple context files coexist, teams may benefit from clearly structuring them hierarchically (e.g., using tool-specific files as adapters that reference a shared core file) to reflect the layering patterns we identified. Third, for recurring, well-defined workflows, Skills offer the potential for richer configuration through scripts and structured resources, but this potential is currently underutilized. In practice, configuration is presently used more as documentation than as automation, suggesting that teams should adopt executable Skills deliberately and only when the expected benefits justify the additional setup effort. Finally, given the distinct configuration profiles across tools, teams should be mindful that adopting tool-specific mechanisms (e.g., rule-based configurations) may shape their workflows in tool-dependent ways.

For researchers, our findings point to several directions. Longitudinal studies are needed to track how configuration practices evolve as tools mature and as developers gain experience. Controlled studies should assess whether advanced mechanisms such as Skills with executable resources or dedicated Subagents provide measurable benefits over Context Files alone, extending prior work on the operational impact of `AGENTS.md` [13]. Given that projects increasingly use multiple tools (Section 5.1), research should also investigate interoperability challenges: how configuration artifacts interact across tools, and how potential conflicts between overlapping or contradictory instructions in different files can be detected and resolved. Our automated data collection and analysis pipeline [5] supports such ongoing analyses.

Finally, our study is limited to a single point-in-time snapshot of a rapidly evolving landscape. Agentic AI coding tools are under active development, and both the available configuration mechanisms and the patterns of their adoption are likely to change. The trends we identify—toward standardization around `AGENTS.md`, shallow adoption of advanced mechanisms, and tool-specific configuration cultures—are early empirical signals rather than settled findings.

## 8  Conclusions

Our study is the first to provide a comprehensive overview of the configuration mechanisms used by agentic AI coding tools in software engineering. We systematically identified eight configuration mechanisms across five agentic AI coding tools and analyzed their adoption in 2,926 GitHub repositories. Three findings stand out. First, Context Files dominate and are often the only mechanism present, with `AGENTS.md` emerging as an interoperable standard. Second, advanced mechanisms such as Skills and Subagents remain shallowly adopted, with Skills used predominantly as static documents rather than executable workflows. Third, distinct configuration cultures are forming around different tools. For practitioners, these findings suggest that `AGENTS.md` is the natural starting point for configuring agentic tools, especially in multi-tool environments, and that Skills offer untapped potential for encoding recurring workflows beyond static instructions. Tool providers should consider improving onboarding and documentation for advanced mechanisms, given the gap between their expressive power and current adoption. For researchers, controlled studies are needed to determine whether richer configuration, such as Skills with executable resources, yields measurable improvements over Context Files alone, and how conflicts between configuration artifacts in multi-tool repositories can be detected and resolved. Longitudinal research needs to track how configuration patterns evolve.

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
