# OpenReview forum: "Configuring Agentic AI Coding Tools: An Exploratory Study"
_ACM.org/AIWare/2026/Conference — AIware 2026_

### Official Review · Reviewer_fHmp · 2026-03-11

**Rating:** 3
**Confidence:** 4

**Review:**

### Strengths
- Timely and clear taxonomy of the configuration mechanisms
- Dataset availability on Zenodo

### Weaknesses
- Lack of deep analysis (e.g., analysis of configuration content or impact)
- No actionable guidance

### Detailed Comments
1. The study detects whether configuration files exist in repositories, but it does not examine what developers actually write inside them. It would be valuable to categorize the content of context files. For example, through topic modeling or manual coding of a representative sample of CLAUDE.md and AGENTS.md files. This would show common instruction patterns and potential anti-patterns to enhance the paper.

2. The study provides a single point in time for configuration files. However, since these files are version controlled, commit-level evolution analysis is feasible with the data already collected. It would strengthen the paper to track when these files are created relative to project milestones, how frequently they are updated, and how their content changes as projects grow.

3. The study reports what configurations exist across repositories, but it does not investigate whether these configurations have any measurable effect on project outcomes.

4. The study reports language-specific adoption patterns, but it does not analyze how configurations differ across project domains (e.g., AI vs. web applications). It would be helpful to group repositories by domain and report configuration patterns per domain. Practitioners would benefit from understanding what configurations are commonly adopted for certain types of projects.

5. The study offers practitioner advice, but the recommendations remain generic (e.g., "use AGENTS.md as a starting point," "Skills offer untapped potential"). It would be more actionable to include concrete examples from the dataset. For example, what does a well-structured AGENTS.md look like? What Skill configurations have developers found useful? Presenting representative good and poor examples would make the findings directly useful to practitioners.

6. The study uses file presence heuristics to detect configuration adoption, but it acknowledges that file presence does not confirm active tool usage. It would strengthen the validity of the results to report a manual validation on a random subset. For example, verify whether detected files are non-trivial (e.g., not empty) and whether the corresponding tool shows evidence of active use (e.g., agent-authored commits). Reporting the precision of the heuristics would increase confidence in the findings.

### Minor Comments
- Table 3 needs statistical clarification. If each tool-specific subset is compared against "All", the samples overlap and standard rank-based tests are not appropriate. If the actual comparison is against the complement set, this should be stated explicitly.
- Figure 3 reports counts larger than the total number of repositories, which is confusing without a clearer explanation.
- There are multiple typos (e.g., "patters", "copilot-insturctions").

**Summary:**

This paper studies how agentic AI coding tools are configured in software repositories. The authors review documentation for five agents (Claude Code, GitHub Copilot, Codex, Cursor, and Gemini) and propose a taxonomy of configuration mechanisms. The authors analyze configuration adoption across 2,926 GitHub repositories in terms of configuration distribution, co-occurrence, and evolution. The main findings are that context files dominate adoption, advanced mechanisms such as Skills and Subagents are relatively uncommon, and AGENTS.md increasingly co-occurs with or is referenced by tool-specific files.

---

> ### Author Response · Authors · 2026-03-20
>
> We thank the reviewer for the thoughtful feedback.
>
> Regarding the lack of deep analysis of configuration content, we agree that examining files would provide valuable insights. Our goal, however, is to provide a large-scale empirical characterization of configuration adoption patterns using observable signals (e.g., file presence, co-occurrence). Analyzing the content of files requires different methods and does not scale easily to our dataset. We therefore position this work as a quantitative understanding of configuration practices, upon which more detailed analyses can build.
>
> Similarly, we agree that a longitudinal analysis of configuration evolution would provide useful insights. While version history is available, incorporating commit-level temporal analysis introduces additional complexity in aligning configuration changes with project milestones and ensuring consistency across repositories. Given the scope of this study, we focus on a cross-sectional view that enables consistent comparison across a large number of repositories.
>
> Regarding the absence of analysis on the impact of configurations on project outcomes, we agree that this is an important question. However, establishing such relationships would require careful control of confounding factors (e.g., project size, contributor base, domain, and development practices), which falls outside our scope. Our focus is on identifying and characterizing configuration mechanisms and their adoption patterns, rather than evaluating their effectiveness.
>
> On domain-specific analysis, we acknowledge that grouping repositories by project domain (e.g., AI, web, systems) could provide useful insights. However, accurately inferring project domain at scale introduces its own potential sources of noise. Our current analysis focuses on language-level differences, which provide a reproducible proxy across repositories.
>
> Regarding actionable guidance, we appreciate the suggestion. While the current paper focuses on aggregate patterns, we agree that presenting selected examples (e.g., well-structured configuration files) would further enhance practitioner utility. We acknowledge that recent work has begun to analyze the content of configuration artifacts, and we position our work as complementary, providing a large-scale structural foundation for these directions.
>
> On the concern about heuristic validity, we agree that file presence does not guarantee active use. Our identification strategy is grounded in tool-specific documentation. This provides a consistent and reproducible basis for large-scale analysis. To strengthen validity, we complement our analysis with additional signals such as the presence of AI-authored commits. Among the 2,895 repositories that contain at least one configuration artifact, 2,079 (71.8%) exhibit AI-authored commits, showing that these configurations are associated with active agent-assisted workflows. AI-authored commits were identified using a heuristic based on commit metadata, including author identity fields, structured git trailers (e.g., co-authored-by, generated-by), and commit message patterns. Details will be in the supplementary material.
>
> We also examined all 7,800 configuration files across the eight artifact types to assess content quality. Only 12 files (0.2%) were empty (e.g., zero-byte files, single newlines) and removed from the dataset. We further validated content quality by identifying all files with ten or fewer lines (435 files total). Also, using Claude Code (Opus 4.6 with high effort), one of the authors reviewed these files and found that 396 contained substantive configuration content (e.g., coding conventions, agent definitions), 51 were reference-only files analyzed separately, and four were borderline cases (e.g., minimal headings or filenames).
>
> In addition, we examined the 49 repositories in which a tool was detected but none of the eight defined configuration mechanisms applied (previously labeled as “No config” in Figure 3, now labeled as  “Other” for clarity). Among these, 28 contained miscellaneous tool-specific artifacts (e.g., style guides, planning documents) captured by “Other”, while 21 relied only on deprecated .cursorrules.
>
> The inspection script, files, and their classification will be in the supplementary material.
>
> Regarding the concern in Table 3 about overlaps when comparing tool-specific subsets against “All,” we clarify that all comparisons are performed against the complement set (i.e., repositories not using the given tool), thereby ensuring non-overlapping samples. Specifically, we compare each tool’s adopters against non-adopters across four metrics (i.e., [5 tools] × [4 metrics] comparisons).
>
> For Figure 3, we clarify that counts may exceed the total number of repositories because a single repository can contain configurations for multiple tools and multiple files.
>
> We have also corrected the identified typos and performed an additional proofreading pass to address minor errors.

---

### Official Review · Reviewer_q1oV · 2026-03-11

**Rating:** 3
**Confidence:** 4

**Review:**

I found the work overall interesting and indeed very timely as agentic AI is becoming increasingly popular. While the topic is very relevant to AIWARE there are a few issues that hinder quite a bit the significance of the results and conclusions.

The obvious limitation of this work (also admitted by the authors) is that the study focuses on the presence of configuration artifacts, which mechanisms exist, which are common, and how often they co-occur. This is quite a shallow analysis.

First, the paper targets Agentic AI coding, but no effort (no even simple heuristic) was given to make sure that the configurations really referred to Agents for code generation. Also, the agentic file might also be there for other reasons rather than AI coding. Indeed, the presence of files such as AGENTS.md does not necessarily imply that developers are using agentic AI for code generation. Such files could also exist for other agent-related purposes within the repository. Since the paper relies on filename and path heuristics, stronger validation is needed to show that the detected artifacts truly correspond to adoption of agentic AI coding tools rather than other uses.

Second, there is no deep analysis on the content of those files and what is the purpose and use of those agents.

There are a lot of missing opportunities in this work. Like analyzing common configuration settings across the repos, whether they share similar configurations or not. What is the quality of these configurations? Also studying in depth how these files evolved (longitudinal analysis) would be quite important. However, I think this is fine, as this work wants to be an exploratory study on Agentic configurations that will support future research in this topic.

Another critical issue is the selection of the github repos. The beginning of Section 3 is very confusing and vague. The paper mentions some criteria for the search and for selecting the repos that are not well justified. I don’t understand why the criteria must have repos created before 1 Jan 2024, and therefore excluding newer repositories. Is it because older repos are more likely to be mature? This is not explained. Also many other selection criteria are not justified. The paper cites [16] for the number of commits and watchers, what are the thresholds and how are they motivated? Overall the section 3 is not very clear, with not enough justification of the design choices for the selection. With better selection criteria (less biased, less narrow) the paper could also gauge a current snapshot on how popular are agentic configurations in Github repos. This feels like a huge missed opportunity.

Overall, the paper suffers from moderate issues but I think it remains an interesting and useful work that sheds light on the adoption and configuration of Agentic AI, and will help, inform, and support the research community in proposing more advanced studies and techniques.

**Summary:**

This paper presents an empirical study on open-source github repositories that use AI Agents. The authors detected them using heuristics based on the presence of certain files (e.g., AGENTS.md). The tools studied were Claude Code, Copilot, Cursor, Gemini, and Codex, and checks which setup files and options are used The paper gives a view of how these AI coding tools are being used in practice.

---

> ### Author Response · Authors · 2026-03-20
>
> We thank the reviewer for the thoughtful and constructive feedback.
>
> Regarding the concern that the study focuses on the presence of configuration artifacts and may therefore provide a shallow view, we would like to clarify that the primary goal of this work is to provide a large-scale empirical characterization of adoption patterns. As such, we intentionally focus on observable and reproducible signals (e.g., file presence, co-occurrence, and structural patterns) that can be consistently measured across thousands of repositories. We acknowledge that this does not capture depth of usage or effectiveness, and we explicitly frame our findings as indicators of adoption rather than direct evidence of how these configurations are used in practice. To further support this interpretation, we detected the presence of AI-authored commits within these repositories. Among the 2,895 repositories that contain at least one configuration artifact, 2,079 (71.8%) exhibit AI-authored commits, providing additional evidence that these projects are engaged in active agent-assisted workflows rather than containing purely inert artifacts,
> AI-authored commits were identified using a heuristic based on commit metadata, including author identity fields, structured git trailers (e.g., co-authored-by, generated-by), and commit message patterns. A commit is attributed if any signal matches, enabling robust detection across attribution styles while minimizing false positives. The implementation and dataset will be made available in the supplementary material.
>
> To further ensure that detected artifacts are not trivial, we performed a content quality check across all configuration files. Only 0.2% were empty and were removed from the dataset, and manual inspection of small files confirmed that all contained meaningful content. This strengthens confidence that the detected artifacts reflect substantive configurations rather than placeholders.
>
> Relatedly, our identification strategy is grounded in tool-specific documentation, where such files are defined as configuration entry points for agentic workflows. While alternative uses are possible, these conventions are sufficiently standardized across the studied tools to serve as a practical proxy for large-scale analysis. Furthermore, our dataset construction focuses on repositories that exhibit multiple independent signals of agent-related configuration (e.g., co-occurring configuration files and tool-specific directories), which reduces the likelihood that detected artifacts are incidental or unrelated. Importantly, our goal is not to estimate absolute adoption rates, but to comparatively analyze configuration patterns across tools under consistent detection criteria.
>
> In the absence of deeper content analysis, we agree that examining the semantics, quality, and intent of configurations would provide valuable additional insights. However, such analysis requires fundamentally different methods (e.g., qualitative coding or semantic interpretation) and does not scale easily to the size of the dataset considered in this work. We therefore position this study as an exploratory, large-scale characterization that complements, rather than replaces, more in-depth qualitative investigations. We have also clarified this positioning and highlighted these opportunities more explicitly in the discussion.
>
> Regarding the selection of GitHub repositories, the criteria were designed to balance dataset quality, relevance, and comparability. In particular, restricting repositories to those created before 1 January 2024 ensures approximately two years of project history at the time of data collection, ensuring the inclusion of repositories with a certain level of maturity. This reduces the influence of very recent repositories where configuration practices may still be evolving, as well as repositories created primarily for experimentation or short-term exploration of AI configuration mechanisms. Similarly, thresholds on commits and watchers were applied to reduce noise from inactive, toy, or auto-generated repositories, which are prevalent due to the highly skewed distribution of activity and popularity on GitHub. Rather than using arbitrary cutoffs, we adopt median-based thresholds following [16] (reference in the paper), providing a data-driven and consistent filtering strategy. Together, these criteria focus the analysis on repositories where configuration practices are more likely to be meaningful, while making the associated trade-offs and potential biases explicit. Alternative dataset constructions (e.g., including newer repositories or relaxing thresholds) could provide complementary perspectives, such as a more current snapshot of adoption. However, our design prioritizes the selection of repositories containing engineered software projects with sufficient history and activity.

---

### Official Review · Reviewer_cPKM · 2026-03-12

**Rating:** 3
**Confidence:** 3

**Review:**

**Strengths:**

- **Timely and relevant topic.** Agentic AI coding tools are rapidly reshaping software development workflows, yet there is little empirical understanding of how developers configure them. This paper fills a genuine gap and addresses a topic squarely within AIWare's scope (Agents & SE, AI-driven software practices).

- **Systematic identification of configuration mechanisms.** The taxonomy of eight mechanisms (Table 1) is a useful contribution. The cross-tool comparison clearly shows where tools converge (e.g., Context Files, Skills) and diverge, providing a practical reference for both researchers and practitioners.

- **Reasonable scale and methodological rigor.** The data collection pipeline (Figure 1) is well-documented, starting from 37,249 repositories and filtering down to 2,926 through a reproducible series of steps. The use of GPT-5.2 for classifying "engineered" projects, combined with spot-checking, is a pragmatic approach. Statistical tests with appropriate corrections (Mann-Whitney U with Benjamini-Hochberg) are applied in Table 3.

- **Rich descriptive analysis.** The paper provides multi-faceted views of adoption: distribution across tools (Figure 3), temporal trends (Figure 4), co-occurrence patterns (Section 5.2.1), creation order sequences (Figure 6), and inter-file reference networks (Figure 7). These complement each other well and paint a coherent picture.

- **Actionable practical implications.** The discussion offers concrete recommendations for practitioners (e.g., adopt AGENTS.md as a cross-tool baseline) and identifies clear future research directions (longitudinal studies, controlled experiments on configuration effectiveness).

**Weaknesses:**

- **Construct validity: presence ≠ active use.** The paper acknowledges this (Section 6.4) but does not sufficiently mitigate it. Detecting a `CLAUDE.md` file tells us the file exists, not that it meaningfully influences agent behavior or developer productivity. Many configuration files may be auto-generated, boilerplate, or abandoned.

- **No validation of classification heuristics.** The heuristics for detecting configuration mechanisms (Table 1, based on file names and directory patterns) are central to the study, yet no precision/recall evaluation is reported. False positives (e.g., a `.claude/` directory used for unrelated purposes) and false negatives (e.g., non-standard file names) could bias results. At minimum, a manual validation on a random sample with inter-rater agreement metrics should be provided.

- **Reliance on a single LLM for repository classification.** Using GPT-5.2 to label repositories as "engineered" is a non-trivial methodological choice. The paper mentions "iterative prompt development and spot-checks" but does not report agreement rates, the number of spot-checked samples, or comparison with human labels.

- **Limited depth in RQ3 analysis.** While RQ3 promises to examine "how" configuration mechanisms are adopted, the analysis remains largely quantitative and structural (file counts, resource type distributions, line counts). Qualitative content analysis, (e.g., what instructions do developers actually write in Context Files? What kinds of tasks do Skills encode?), would significantly deepen the contribution. The brief mention of three reference patterns (Section 6.1) is a positive step but feels underdeveloped.

- **Threats from multi-tool overlap.** Repositories using multiple tools (e.g., Claude + Copilot) may conflate tool-specific adoption patterns. The paper notes this (Section 5.1) but does not control for it in the analysis. Separating single-tool repositories from multi-tool ones in key analyses would clarify tool-specific trends.

**Summary:**

This paper presents a systematic analysis of configuration mechanisms for agentic AI coding tools (Claude Code, GitHub Copilot, Cursor, Gemini CLI, and Codex CLI). The authors identify eight configuration mechanisms (Context Files, Skills, Subagents, Commands, Rules, Settings, Hooks, and MCP) through documentation review, then conduct an empirical study of 2,926 GitHub repositories to examine adoption patterns. Three key findings emerge: (1) Context Files dominate the configuration landscape, with AGENTS.md converging as an interoperable standard; (2) advanced mechanisms like Skills and Subagents see only shallow adoption, relying on static documentation rather than executable workflows; (3) distinct tool-specific configuration cultures are forming, with Claude Code users employing the broadest range of mechanisms.

---

> ### Author Response · Authors · 2026-03-20
>
> We thank the reviewer for the insightful and constructive feedback.
>
> Regarding construct validity, we agree that the presence of configuration artifacts does not necessarily imply active or effective use. We have been careful to frame our findings accordingly, particularly in the threats to validity section. To complement this, we examined evidence of AI-assisted activity within repositories through the presence of AI-authored commits. Among the 2,895 repositories that contain at least one configuration artifact, 2,079 (71.8%) exhibit AI-authored commits, providing additional evidence that these configurations are associated with active agent-assisted workflows.
>
> AI-authored commits were identified using a heuristic based on commit metadata, including author identity fields, structured git trailers (e.g., co-authored-by, generated-by), and commit message patterns. A commit is attributed if any signal matches, enabling robust detection across attribution styles while minimizing false positives.
>
> We also examined all 7,800 configuration files across the eight artifact types to assess content quality. Only 12 files (0.2%) were empty (e.g., zero-byte files, single newlines); these were removed from the dataset. We further validated content quality by programmatically identifying all files with ten or fewer lines (435 files total) and conducting a detailed inspection. Using Claude Code (Opus 4.6 with high effort), one of the authors reviewed these files and found that 396 contained substantive configuration content (e.g., coding conventions, agent definitions, and tool settings), 51 were reference-only files analyzed separately in Section 6, and four were borderline cases (e.g., minimal headings or filenames).
>
> On the validation of classification heuristics, our approach is grounded in the official documentation of each tool, which defines canonical file names, directory structures, and configuration conventions. These conventions are explicitly designed by tool developers to standardize usage, and thus provide a consistent and tool-aligned basis for large-scale identification. While we acknowledge that edge cases may exist (e.g., unconventional naming or repurposed directories), such cases are unlikely to systematically bias the results given the scale of the dataset.
>
> To further assess potential false positives in our heuristics, we manually re-checked the 49 repositories (Figure 3) where a tool was detected without any of the specified configuration mechanisms. This subset represents the most likely source of false positives. Of these, 28 contained miscellaneous tool-specific artifacts (e.g., style guides, planning documents, or ignore files) captured under the “Other” category (previously labeled as “No config” which we have renamed for clarity), while 21 relied exclusively on the deprecated .cursorrules format. This suggests that such cases are limited and primarily reflect how tool-to-configuration mappings are defined (i.e., which files or directories are considered valid configuration artifacts), rather than systematic misclassification.
>
> Regarding the classification of repositories based on one LLM, we classified repositories to focus the dataset on engineered repositories. While this classification relied on a single LLM, this should not directly affect the detection of configuration mechanisms, which is independent of the LLM for classification. Regarding the single LLM for classifying repositories, we experimented with multiple GPT models including default gpt-5.2, gpt-5-nano, and gpt-5-mini and selected GPT-5.2 based on its superior performance. The model was applied with a deliberately conservative strategy. Repositories for which the model returned an “unsure” label were excluded from the final dataset. This exclusion strengthens rather than weakens our findings, as it reduces the risk of false positives in the retained set.
>
> With respect to the depth of RQ3, the intent is to characterize how configuration mechanisms are adopted at scale through observable structural patterns (e.g., file types, co-occurrence, sequencing, and references). While a qualitative analysis of configuration files would provide complementary insights, it lies outside the scope of this study. We therefore position this work as establishing a quantitative foundation upon which more detailed qualitative analyses can build in the future.
>
> Regarding potential confounding from multi-tool repositories, we focus on the presence and distribution of configuration mechanisms rather than attributing causal effects to individual tools. Multi-tool usage reflects the practical reality of developers. While such overlap may influence downstream behavioral outcomes, it does not invalidate the descriptive analysis of configuration adoption patterns. We now discuss this explicitly, noting that isolating tool-specific effects would require a different study design.